# "Research Mentors Should Support Students of Color by Taking an Extra Step." Undergraduates' Reasoning about Race and STEM Research Mentorship

**Kristin Dee Vierra** [1,*], **Diana R. Beltran** [1] , **Lindsay Denecker** [2] **and Rachael D. Robnett** [1]

1   Department of Psychology, University of Nevada, Las Vegas, NV 89154, USA;
gutied14@unlv.nevada.edu (D.R.B.); rachael.robnett@unlv.edu (R.D.R.)

2   Department of Psychology, Bowling Green State University, Bowling Green, OH 43403, USA;
ldeneck@bgsu.edu

\*   Correspondence: kristindvierra@gmail.com

**Abstract:** Racial inequities and the adoption of a colorblind approach in education contribute to a situation wherein many academics lack the personal experience and incentive needed to identify and confront racism within society and institutions. This may be particularly the case in fields related to science, technology, engineering, and math (STEM), which tend to have lower levels of diversity compared to other fields. The current study examines undergraduates' perceptions of whether STEM research mentors should consider race when mentoring students from marginalized backgrounds. We employ a mixed-methods approach with the goal of uncovering how undergraduate students perceive and rationalize research mentoring practices. Findings reveal that a strong majority of undergraduate students believe that research mentors should take race into account when mentoring students from marginalized backgrounds. During the qualitative coding process, the research team unearthed seven overarching themes that outline undergraduates' reasoning, both in favor and against STEM research mentors considering race. We conclude by presenting an intervention intended to motivate individuals to redress colorblind ideologies and thus support a culturally sensitive mentoring style.

**Keywords:** STEM education; colorblind racial attitudes; higher education; STEM research mentorship; culturally sensitive mentoring; color-conscious

## 1. Introduction

The persistent shortage of people with the training and desire to enter the STEM workforce is a serious economic concern within the U.S. [1]. This shortage is compounded by the fact that many STEM fields continue to exhibit substantial racial inequities [2–6]. Failing to recruit and retain members of marginalized racial groups hinders STEM innovation and constitutes a social justice problem. Because STEM careers tend to be lucrative and prestigious, uneven racial representation in these careers contributes to existing structural inequities.

In an effort to broaden and diversify the STEM workforce, a variety of federally funded programs provide undergraduates with resources that enable them to conduct hands-on scientific research under the guidance of more experienced research mentors. Indeed, research mentoring is arguably one of the most pervasive strategies for bringing undergraduates from marginalized backgrounds into STEM careers (for a review, see [7]). Importantly, Students of Color report a range of experiences—some positive and some negative—within the context of their mentoring relationships [8–10]. More so, a growing body of evidence indicates that some research mentors intentionally attempt to ignore race and racial inequity (e.g., a colorblind approach) when working with Students of Color [2–4,11,12].

As discussed below, indirect evidence suggests that colorblind approaches to mentoring can be harmful to Students of Color, given that these approaches disregard the

structural barriers and biases that continue to perpetuate disparities in STEM education and careers [3,11,13]. Surprisingly, however, very little research has directly asked undergraduates whether they think research mentors should acknowledge race when working with Students of Color. Foregrounding undergraduates' perspectives is crucial, given that they are key stakeholders in research mentoring relationships. Moreover, their perspectives will also provide insight into whether colorblind ideologies that faculty espouse "trickle down" to undergraduates and inform how they think about diversity in higher education [14].

To provide insight into these issues, the current research leveraged a unique sample of undergraduates who (a) attended one of the most racially diverse universities in the U.S. and (b) reported prior experience with mentoring to provide insight into these issues. Consistent with the National Science Foundation (NSF) definition of STEM [15], participants were from a range of academic disciplines that included the so-called bench sciences (e.g., biology and chemistry) as well as social science fields (e.g., psychology). The study employed a mixed-methods approach. Specifically, we asked a sample of undergraduates to reflect on how research mentors should approach mentoring relationships with Students of Color. We were particularly interested in whether participants would use colorblind ideologies to justify the belief that mentors should refrain from taking race into account in their mentoring relationships. We also tested for quantitative associations between participants' qualitative reasoning and their level of colorblind ideology as well as their sociodemographic background.

### 1.1. STEM Research Mentoring

Mentoring is an interactive and mutually beneficial learning relationship that aims to aid mentees in developing the fundamental skills and knowledge necessary for succeeding in their preferred careers [16]. Scholars suggest that quality mentoring is critical for college students' success [17–19]. This may be particularly the case for students from marginalized backgrounds, given that many of these students are underrepresented in their academic programs and encounter both subtle and overt bias in academic settings [20,21]. Students interested in pursuing a STEM career and/or furthering their education at the graduate level will commonly work with a research mentor [22]. Similar to more general forms of academic and career mentoring, STEM research mentoring is a mutually beneficial relationship in which a less experienced student learns the practice of scientific research by working under the guidance of a more experienced mentor [7]. Research mentors play a pivotal role in influencing trainees' research career intentions, academic persistence, self-efficacy, academic identity, and a range of other attributes associated with academic success [7,23–27]. For this reason, research mentoring is the cornerstone of many academic outreach programs (e.g., McNair) that seek to redress sociodemographic inequities in STEM fields.

Importantly, however, not all mentoring relationships are created equal. Although some serve their students well, others do not. For example, past research indicates that research mentors employ colorblind ideologies when working with students from marginalized racial backgrounds [11,28–30]. Colorblind ideologies are defined as the denial or belittlement of race and racism [31,32]. Further, McCoy and colleagues [11] found that research mentors who employ colorblind ideologies described students from marginalized backgrounds as academically inferior and less prepared or interested in pursuing research compared to their White counterparts.

Colorblind perspectives are concerning for several reasons. For example, they may lead STEM research mentors to undermine the potential and restrict opportunities for students from marginalized racial backgrounds, further perpetuating disparities in STEM education. Consistent with this point, research suggests that students benefit when their mentors are culturally sensitive. Mentors who are culturally sensitive strive to be cognizant of their mentees' racial and ethnic backgrounds (and corresponding systems of oppression and privilege) and adjust their mentorship strategies as appropriate [13,20,33–35]. Examples

of culturally sensitive mentoring include creating a safe and inclusive space where students feel comfortable turning to their mentor for guidance and resisting negative stereotypes about students from marginalized backgrounds [20,36]. One study found that when mentors adopted a culturally sensitive mentorship ideology, their mentees were more likely to rate their mentoring relationship as favorable, identify clear academic and career goals, and report feeling competent as a researcher [37].

Despite culturally sensitive mentoring being a best practice, it is not clear how receptive undergraduates are to this practice. Investigating this question is important for several reasons. First, from a social justice standpoint, understanding how undergraduates reason about equity-focused initiatives (research mentoring, in this case) is a necessary component of creating campus climates that are more inclusive. Second, and relatedly, undergraduates are currently in the developmental stage of emerging adulthood. Emerging adulthood is a developmental period that typically unfolds in the third decade of life [38]. During emerging adulthood, young people are in the midst of exploring and consolidating their worldviews while also making crucial decisions about their futures [38,39]. Consequently, the ways in which emerging adults reason about societal issues lay the foundation for their subsequent engagement and approach to addressing societal problems later in life. Third, understanding emerging adults' attitudes about outreach initiatives such as research mentoring is central to understanding the racial ideologies that characterize the broader university landscape.

### 1.2. Colorblindness

In the United States, a pervasive narrative holds that American society has entered a post-racial era wherein the relevance of race and the existence of racism have become obsolete [40]. In line with this narrative, many Americans adopt a colorblind ideology that eschews the acknowledgment of race and racism and, instead, emphasizes sameness and equality across racial groups. Individuals who adopt a colorblind ideology avoid defining, framing, or pathologizing Whiteness, which perpetuates racism and maintains White supremacy [41]. A colorblind ideology posits that to achieve racial equality, individuals should be treated without regard to their racial identity, yet this lens overlooks the historical injustices and systemic inequalities faced by minoritized groups [31,32,40]. In short, colorblind ideologies are defined as the denial or belittlement of race and racism [31,32].

Colorblind ideologies can be classified into two categories [42–44]. The first category, known as power evasion, is defined as a failure to recognize racial privilege and institutional racism [43]. For example, an individual subscribing to a power evasion viewpoint may argue that society is purely meritocratic, and everyone has an equal opportunity to succeed, regardless of their race. However, this individual fails to acknowledge the historical and structural barriers that have disadvantaged People of Color. The second category is color evasion, marked by the declaration of not "seeing race" [43]. For example, an individual might state, "I don't see color. We're all the same". This statement is problematic because it overlooks the fact that People of Color often have different experiences and face distinct challenges due to structural racism.

Not surprisingly, there is racial variation in the degree to which people endorse colorblind ideologies. Colorblind ideologies play a role in upholding White supremacy by promoting a refusal to acknowledge racial privilege and structural racism [32,41,45]. Therefore, compared to members of marginalized ethnic groups, White individuals tend to endorse higher levels of colorblind ideology, which enables them to preserve and perpetuate their social and systemic privileges [32,46]. Colorblind ideologies have also been documented in Asian American samples, which is likely related in part to the model minority myth [47–50]. The model minority myth is a false stereotype that characterizes Asian American individuals as more academically, economically, and socially prosperous than other racial minority groups [51–55]. Endorsing the model minority myth may foster a sense of proximity to Whiteness and corresponding colorblind ideologies among some

Asian American people [49,53,56,57]. These general trends informed the ethnic comparisons that were carried out in the current research.

In educational contexts, the concept of colorblindness is commonly endorsed and can influence policy and social practices regarding diversity. Many college educators believe it is best to ignore the race or ethnicity of their students, which they justify by drawing from their purported commitment to treating all students equally [11,58]. However, when faculty members attempt to treat students equally, they ignore the systemic barriers students from marginalized backgrounds experience and fail to celebrate the cultural assets that these students bring into academic contexts [11,13].

Racial bias in STEM is often observed through the perspectives of colorblindness [59,60]. Despite being portrayed as a competitive and meritocratic domain, there is evidence indicating a higher prevalence of racial bias within STEM professions compared to non-STEM fields [56]. Moreover, many STEM students from marginalized backgrounds encounter instances of racial microaggressions and stereotyping [61–63]. Further, scholars note that STEM faculty with stronger endorsements of colorblind ideologies report reduced application of inclusive teaching techniques [64]. These faculty members tend to ascribe to the belief that race is unrelated to the learning environment and often perceive themselves as immune from cultural subjectivity or implicit bias [28,30,65].

## 2. Materials and Methods

The work described thus far suggests that colorblind ideologies are common in STEM research mentoring relationships and on college campuses more generally [11,44,58,64,66]. Although evidence suggests that these ideologies can be harmful, limited research has directly asked undergraduates to reflect on how research mentors should relate to students from marginalized racial backgrounds. To this end, the current study explored whether undergraduates think research mentors should acknowledge race when mentoring undergraduates from marginalized racial backgrounds. This leads us to

**Research Question 1:** *Do undergraduates think research mentors should take race into account when mentoring students from marginalized racial backgrounds? How do they explain their reasoning?*

We suspect that one influential factor shaping undergraduates' reasoning is colorblind ideology. Therefore, in addition to inductive (i.e., exploratory) qualitative analyses, we also employed a deductive (i.e., theory-driven) qualitative approach to examine whether students who believe research mentors should not take race into account use colorblind ideologies to justify their view.

Next, we delve more deeply into the qualitative data to explore ethnic differences in the patterns that emerged in the analyses pertaining to Research Question 1. Guided by the scholarship described earlier, [49,53,56,57] we examined whether the reasoning that White and Asian American students provided was different from the reasoning that Black and Hispanic/Latinx students provided. Specifically, Research Question 2 is as follows:

**Research Question 2:** *Compared to Asian American and White students, do Black and Hispanic/Latinx students differ in their reasoning about whether research mentors should take race into account when mentoring students from marginalized backgrounds?*

After completing the qualitative analyses, we conducted quantitative analyses to determine whether undergraduates' ethnic-racial background is associated with their perception of whether research mentors should take race into account when mentoring students from marginalized racial backgrounds. As mentioned above, White and Asian American individuals tend to endorse higher levels of colorblind ideologies [32,46,49,56,57]. Therefore, Hypothesis 1 is as follows:

**Hypothesis 1:** *Black and Hispanic/Latinx students will be more likely to say that mentors should take race into account when mentoring students from marginalized backgrounds compared to White and Asian American students.*

We were also interested in exploring whether other aspects of sociodemographic backgrounds were associated with how participants reason about mentors' approach to addressing race in their mentoring relationships. Accordingly, Research Question 3 is as follows:

**Research Question 3:** *Do beliefs about whether mentors should take race into account in their mentoring relationships differ on the basis of participants' gender, political party affiliation, first-generation status, and scores on a quantitative scale measuring colorblind ideology?*

We address these hypotheses and research questions through a combination of qualitative and quantitative data (i.e., a mixed-methods approach). A mixed-methods approach offers several advantages compared to relying solely on qualitative or quantitative methods. For instance, it allows for an in-depth understanding of how participants approach a specific question, going beyond mere agreement or disagreement [67]. Moreover, it capitalizes on the strengths of both quantitative and qualitative data, facilitating the exploration of a broader and more intricate spectrum of issues [68].

*2.1. Participants*

Participants were recruited from a public research university located in the southwestern region of the U.S. The university is consistently ranked as one of the most racially diverse campuses in the U.S., which is evidenced by its status as a Minority-Serving Institution, Hispanic-Serving Institution, and Asian American and Pacific Islander-Serving Institution. The economic background of students at this university closely aligns with that of students attending similarly selective public colleges in the same region and throughout the country [69].

Participants were asked to complete an online survey for course credit during the 2022 academic year. Although participants were part of a psychology participant pool, they represented a wide range of academic disciplines: Specifically, about half of the participants were from "bench-science" STEM fields (e.g., chemistry and biology) and about half were from social science STEM fields (e.g., psychology) (The academic diversity in our sample is unsurprising. At the university where data collection occurred, psychology is a common general education requirement across a number of majors. Further, psychology content was recently added to the MCAT (a standardized test for admission into medical school), which has led to an influx of pre-med students into psychology courses.). It is important to note that our study focused on a distinct subset of the larger participant pool. Specifically, the current research focuses on 216 undergraduates who reported having experience with mentoring.

The majority of participants (93%) were between the ages of 18 and 24. The sample included 147 women (68.1%), 63 men (29.2%), 1 transgender woman (0.5%), 2 participants who identified as gender nonconforming (0.9%), 2 participants who identified as nonbinary (0.9%), and 1 participant who elected not to disclose their gender (0.5%). Regrettably, participants who did not identify as either women or men were excluded from the quantitative gender comparisons due to limitations imposed by the sample size. With regard to political ideology, 94 participants (43.5%) identified as Democrat, 64 participants (29.6%) identified as "Other" or Libertarian, 30 participants (13.9%) identified as Republican, 25 participants (11.6%) identified as Progressive or Socialist, and 3 participants chose not to specify.

With respect to ethnic-racial background, 54 participants (25.0%) identified as Hispanic or Latinx, 48 participants (22.2%) identified as Asian American or Pacific Islander, 46 participants (21.3%) identified as White, 23 participants (10.6%) identified as Black or African American, and 45 participants (20.8%) identified as a member of a different ethnic group. Participants who identified as both White and a member of a marginalized group

were categorized as members of the marginalized group. For instance, if a participant identified as both White and Black, they were classified as Black. Participants were also asked what race most people would perceive them to be (i.e., street race; [70]) to assess the accuracy of our self-reported ethnic identity measure. The percentages obtained from this street race measure were largely consistent with the percentages reported earlier. However, a slight discrepancy was noted among White participants: 33 percent indicated that people would perceive them as White, whereas our self-report measure categorized 21 percent of participants as White. A full description of demographic information can be found in Table 1.

**Table 1.** Demographic Data.

| Demographics | n | % |
|---|---|---|
| **Gender** | | |
| Man | 63 | 29.2 |
| Woman | 147 | 147 |
| Transgender woman | 1 | 0.5 |
| Gender nonconforming | 2 | 0.9 |
| Nonbinary | 2 | 0.9 |
| **Race** | | |
| Black/African American | 23 | 10.6 |
| Asian/Asian American | 48 | 22.2 |
| Hispanic/Latinx | 54 | 25.0 |
| White | 46 | 21.3 |
| Multi-racial | 45 | 20.8 |
| **Political** | | |
| Democrat | 94 | 43.5 |
| Progressive + Socialist | 25 | 11.6 |
| Republican | 30 | 13.9 |
| Libertarian + Other | 64 | 29.6 |
| **Street race** | | |
| White | 73 | 33.8 |
| Black | 26 | 12.0 |
| Asian | 62 | 28.7 |
| Hispanic/Latinx | 50 | 23.1 |
| Other | 4 | 1.9 |

*2.2. Procedure*

Prior to beginning the survey, all participants provided their consent to participate in the study. The survey included a short demographic questionnaire, a quantitative measure assessing participants' colorblindness, and an open-ended question gauging participants' perceptions of whether research mentors should consider race when mentoring students from marginalized backgrounds.

*2.3. Measures*

2.3.1. Racial Colorblindness

The colorblind racial attitudes scale is a 20-item instrument designed to measure colorblind racial attitudes [32]. For example, participants responded to items such as, "Racial problems in the U.S. are rare, isolated situations". Participants responded on a scale ranging from 1 (*strongly disagree*) to 7 (*strongly agree*). Higher scores reflect higher colorblind racial attitudes. Scores on this measure demonstrated adequate internal reliability ($\alpha = 0.77$).

2.3.2. Demographics

Participants were asked to share their ethnicity, street race, gender, education level, socio-economic status, and political affiliation.

*2.4. Qualitative Coding and Analysis*

To investigate how undergraduate students reason about whether research mentors should take race into account when mentoring students from marginalized backgrounds (Research Question 1), we employed an open-ended question format. First, participants were asked, "When research mentors work with Students of Color, should they be doing anything in particular to support Students of Color?" Participants were presented with both a "yes" and a "no" option to choose from. Subsequently, participants were asked to provide their reasoning based on their chosen responses. Participants who selected "yes" were prompted with the following question: "*Please explain why you believe that research mentors who work with Students of Color **should** be doing something to support Students of Color*?" Conversely, participants who selected "no" were asked the following: "*Please explain why you believe that research mentors who work with Students of Color **should not** be doing something to support Students of Color*?"

The open-ended data were analyzed using thematic analysis, which is a qualitative method aimed at identifying, analyzing, and reporting themes within the data. The analytic approach followed [71] recommendations for thematic analysis. Initially, the lead author conducted a comprehensive reading of the entire dataset and designed a coding manual. This manual was developed through a combination of deductive (theory-driven) and inductive (data-driven) methods. More specifically, we utilized a deductive approach, guided by prior scholarship that has centered on the concept of colorblind ideology [43], to assess and compare the prevalence of the two categories of colorblindness: power evasion and color evasion. The coding manual included seven overarching themes. To assess inter-rater reliability, three of the current study's authors and two undergraduate research assistants independently (i.e., separately) used the coding manual to code all 216 responses. Regular meetings were held throughout the coding process to calculate inter-rater reliability and check for coder drift. Inter-rater reliability, which was indexed by Cohen's kappa, was good-to-excellent throughout the coding process ($k = 0.78$ to $0.93$).

To report and interpret the identified coding categories, the research team employed a blended epistemological approach that incorporated elements of both essentialism/realism and a constructionist perspective [71]. An essentialism/realism approach includes formulating theories about motivations and meaning beyond what is said in the qualitative responses. On the other hand, a constructivist approach does not seek to identify motivations and meaning and, instead, simply lists patterns identified in the data. All identified themes, coding categories, and sample responses are presented in Table 2.

**Table 2.** Qualitative themes.

| Race-Consciousness Themes | | | |
|---|---|---|---|
| Theme | Definition | Example | Frequency |
| Culturally sensitive | Participants stress that research mentors should create an inclusive and supportive environment for students of color to feel safe, valued, empowered, and supported. Moreover, they underscore the importance of mentors embracing diverse perspectives and understanding the unique backgrounds of students of color. | "Students of color deserve reassurance and appreciation given the discriminatory circumstances they face." (Black student; gender not specified) | 51 |

**Table 2.** *Cont.*

| Race-Consciousness Themes | | | |
|---|---|---|---|
| Acknowledge adversity | Participants emphasize the importance of research mentors recognizing the adversities faced by students of color, including discrimination, limited resources, disparagement, heightened expectations, stress, and mistreatment. Additionally, they advocate for providing additional support to acknowledge and address these hardships. | "Students of color are constantly mistreated by the people surrounding them so I believe mentors should be there for them always especially in their times of need." (Hispanic or Latinx woman) | 48 |
| Negative behaviors | Participants assert that research mentors should refrain from engaging in certain negative behaviors. Specifically, they should avoid pretending to understand without genuine knowledge, or presuming to possess comprehensive understanding of a student of color's background. | "They should not always say they fully understand or try to manipulate the conversation." (Multi-racial woman) | 6 |
| *Race-Ignoring Themes* | | | |
| Power Evasion | Participants lack an awareness of racial privilege and institutional discrimination. For example, participants state that everyone should be treated "equally" or "the same." | "No matter the color of the student I believe that you should treat everyone the same." (Hispanic or Latinx Woman) | 39 |
| Favoring | Participants maintain the belief that research mentors should refrain from taking specific actions to support students of color to prevent the perception of preferential treatment, undue advantages, or favoritism. | "Students want to feel like students, not as if they need special treatment in order to accomplish the same tasks their peers do" (White male) | 8 |
| Condescending | Participants explain that research mentors should avoid making students of color feel singled out or uncomfortable, as it may create a sense of being treated differently. Additionally, some participants express that mentors should refrain from taking specific actions that might convey a sense of pity towards students of color. | "When research mentors treat students of color differently it starts to feel like mentor feels sorry for them..." (Multi-racial woman) | 8 |

**Table 2.** *Cont.*

| Race-Consciousness Themes | | | |
|---|---|---|---|
| Individualized | Participants express that research mentors should consider all aspects of a student's identity, such as gender, sexual orientation, and socio-economic status, when considering support for students of color. | "I think that research mentors should have an open mind and consider all factors with all students and do the best for all students despite what color the student is." (White woman) | 5 |
| Color Evasion | Participants state that they "do not see color" | "Because i don't see color."(Black woman) | 1 |

### 2.5. Researcher Positionality

During the process of data analysis and coding, the research team actively engaged in self-reflection and extensive discussions regarding the potential influence of our individual positionality on the formulation of research questions and subsequent data analysis. The primary author, a PhD in her twenties, identifies as a White, cisgender, heterosexual woman. The second author, a doctoral student in her twenties, identifies as a Hispanic, cisgender, heterosexual woman. The third author, a doctoral student in her forties, identifies as a White, cisgender, heterosexual woman. The fourth author, an associate professor in her thirties, identifies as a White, cisgender, heterosexual woman. All three authors possess academic training in developmental psychology and the psychology of gender. The remaining team members are psychology undergraduate students who worked as research assistants in the lab where the study took place: (1) A woman in her twenties who identifies as Hispanic or Latinx and heterosexual; (2) A woman in her twenties who identifies as Hispanic or Latino and heterosexual.

### 3. Results

The results will be presented in two sections. First, we present the qualitative results that correspond with Research Questions 1 and 2. Together, these research questions provide insight into (1) whether undergraduates believe that research mentors should take race into account when mentoring students from marginalized backgrounds, (2) the underlying rationale guiding undergraduates in their beliefs, and (3) any potential discrepancies in these beliefs between Asian American and White students in comparison to Black and Hispanic/Latinx students. Next, we present the quantitative findings that correspond with Hypothesis 1 and Research Question 3. Together, this hypothesis and research question provides insight into whether variations in beliefs about whether research mentors should take race into account can be attributed to participants' demographic characteristics or their quantitative levels of colorblindness.

### 3.1. Qualitative

We began by examining how participants reasoned about whether research mentors should consider race when mentoring students from marginalized racial backgrounds. When asked whether research mentors should do anything in particular to support Students of Color, 151 participants (70%) selected "yes". This subgroup was titled *race-conscious mentoring*. The remaining 65 participants (30%) selected "no". This subgroup was titled *race-ignoring mentoring*. Table 3 presents the demographics of each subgroup. The qualitative findings below are organized according to whether participants fell into the *race-conscious or race-ignoring group*.

**Table 3.** Chi-square testing for demographic differences in attitudes about whether research mentors should consider race.

| Demographics | Yes (%) | No (%) | *n* | $\chi^2$ | *df* | *p* |
|---|---|---|---|---|---|---|
| Political | | | | | | |
| Democrat | 77.7 | 22.3 | 210 | 10.21 | 3 | 0.017 * |
| Libertarian + Other | 60.7 | 39.3 | | | | |
| Republican | 53.3 | 46.7 | | | | |
| Progressive + Socialist | 80.0 | 20.0 | | | | |
| Race | | | | | | |
| White and Asian American | 67.7 | 32.3 | 170 | 0.043 | 1 | 0.885 |
| Black and Hispanic and Latinx | 66.2 | 33.8 | | | | |
| Gender | | | | | | |
| Woman | 70.5 | 29.5 | 146 | 0.87 | 1 | 0.350 |
| Man | 63.9 | 36.1 | | | | |
| First-generation status | | | | | | |
| First generation | 72.5 | 27.5 | 213 | 0.77 | 2 | 0.681 |
| Not first generation | 67.6 | 32.4 | | | | |
| Unsure | 63.6 | 36.4 | | | | |
| * $p < 0.05$ | | | | | | |

*3.2. Race-Conscious Mentoring Themes*

Within the race-conscious mentoring subgroup, three primary themes emerged from the data analysis. The first theme was titled *culturally sensitive* (*f* = 51). The second theme was titled, *acknowledge adversity* (*f* = 48). Lastly, the third theme was titled *negative behaviors* (*f* = 6). Although the *negative behaviors* theme was less prevalent in the data, we chose to retain this theme because it contributes to a more comprehensive understanding of *how* participants believe that research mentors should support Students of Color. It is important to note that participants had the flexibility to reference multiple themes in their responses, as these themes were not mutually exclusive. Some qualitative responses (*f* = 55) were not coded due to participants providing rationales that were unclear or too brief to code.

3.2.1. Culturally Sensitive

The first theme encompasses participants' justifications for why they believe research mentors should take action to support Students of Color. A significant number of participants emphasized the importance of research mentors providing a safe and encouraging environment for students while also emphasizing the need for mentors to actively seek an understanding of diverse perspectives and the unique backgrounds of Students of Color. For example, an Asian American woman stated:

> Research mentors [...] should support students of color by taking an extra step. There can be cultural, language, and generational differences that can affect the way a mentor offers advice. Having a background on these differences can make or break the advice that mentors offer.

Furthermore, another participant, a White woman, highlighted the importance of mentors providing a safe environment, stating: "I feel like going out of their way to show their support can help make Students of Color feel welcomed, feel seen". Lastly, an Asian American woman stressed the importance of mentors encouraging Students of Color, stating, "Students of color. . .deserve to have someone that gives them encouragement and motivation". These quotes exemplify the perspective of some students who believe that effective research mentors should prioritize active listening, cultural understanding, and genuine support. These qualities are seen as crucial elements in providing meaningful and impactful mentorship to Students of Color.

### 3.2.2. Acknowledge Adversity

The second theme centers around the challenges faced by Students of Color, with numerous participants emphasizing the importance of research mentors fostering inclusivity and acknowledging the adversities encountered when building a mentoring relationship. Drawing from personal experience, a Black woman shared:

Students of color are always going to have a different experience within college for multiple reasons such as their cultural upbringing or background. College is a privilege for most students of colors, and we don't always have the same resources or knowledge that everyone else has.

In addition, another participant, a White woman, expressed:

I believe students of color often feel a certain way due to biases made against them, racism, etc. It is important for research mentors to keep in mind how past experiences, either individual or culturally, could have affected someone of color.

Lastly, an Asian American man explains:

Research mentors should do something to support students of color because students of color are inherently disadvantaged compared to white students. Students of color have to navigate through a disadvantaged lens of being subject to racial discrimination not only from society and their peers but systemically from colleges and other organizations simply because of their race.

These quotes underscore the view of students who believe that research mentors should be mindful of the disparities faced by students from diverse cultural backgrounds, thereby fostering a supportive and inclusive environment for Students of Color.

### 3.2.3. Negative Behaviors

The third overarching theme addresses behaviors exhibited by research mentors that may seem positive but can negatively impact the mentor-student relationship. This theme addresses the importance of research mentors taking race into account but also emphasizes the importance of ensuring that this consideration is undertaken effectively and meaningfully. In particular, participants voiced concerns regarding mentors pretending to hold knowledge or expertise that they do not genuinely possess or presuming a comprehensive understanding of a student's background when such understanding may, in reality, be lacking. For example, a multi-racial woman stated, "They should not always say they fully understand or try to manipulate the conversation". Further, a multi-racial woman explained that research mentors should avoid "faking the portrayal that they are an ally". These quotes reflect the viewpoint of some students who prioritize inclusive and unbiased mentorship, emphasizing the need for mentors to avoid pretending to know everything and to genuinely understand and appreciate the diverse experiences of Students of Color.

### *3.3. Race-Ignoring Mentoring Themes*

Within the race-ignoring *mentoring group,* four primary themes emerged from the data analysis. The first overarching theme, *colorblind*, was composed of two subthemes: (1) *color evasion* (*f* = 1) and (2) *power evasion* (*f* = 39). The second theme was titled *individualized* (*f* = 5). The third theme was titled *condescending* (*f* = 8). Lastly, the fourth theme was titled *favoring* (*f* = 8). As before, we retained several infrequently occurring themes and subthemes because they were of substantive theoretical or applied importance. Furthermore, participants were able to reference multiple themes in their responses, as the themes were not mutually exclusive. Some qualitative responses (*f* = 15) were not coded due to participants providing rationales that were unclear or too brief to code.

### 3.3.1. Colorblind

The first theme explores colorblind racial attitudes, utilizing a deductive approach to examine the two categories of colorblindness: power and color evasion. The subthemes representing these categories are outlined below. Notably, most responses were classified

under the power evasion category, with a relatively low number falling into the color evasion category.

### 3.3.2. Color Evasion

The first subtheme, color evasion, involves participants adopting a perspective of being "colorblind", where they claim not to see color. Although only a single participant adopted a color evasion approach, we opted to include this subtheme in the analysis to enable a comparative assessment of the prevalence of the two colorblindness categories. The participant who used a color evasion approach, a Black woman, stated, "I don't see color. I feel like everyone is human and goes through the same thing and should be treated the same". This quote reflects that this student believes in treating all students equally, regardless of their racial or ethnic background. Moreso, this quote reflects an emphasis on equality rather than equity.

### 3.3.3. Power Evasion

The second subtheme, power evasion, emerges as participants display a lack of awareness regarding racial privilege and the institutional structures that contribute to inequality. One participant, a White man, reflected on his own mentorship relationship and the advice given by his mentor, stating, "My mentor always said it doesn't matter your race, ethnic background, personal background, genetics... we can always achieve what we want with the right work ethic and dedication". Further, a Latina woman expressed her belief in treating all students equally, stating, "I believe that you should treat everyone the same. It doesn't matter if they are a different color, they are a student". Additionally, a White man stated, "I think that all students who work with mentors should be treated/taught the same way. Teach everybody academics the same way". Another participant, a White woman, reflected on both Students of Color and White students, suggesting that they are both capable of achieving what they want, stating:

> I believe that both people of color and White people have the same intellectual abilities, and do not require different treatment in the academic environment just because of the color of their skin. Both are perfectly capable of achieving the same achievements.

These quotes reflect that some students may downplay the influence of racial privilege and systemic barriers in mentorship and academic environments.

### 3.3.4. Individualized

The second theme highlights the importance of research mentors considering all aspects of a student's identity, including gender, sexual orientation, and socio-economic status when providing support for Students of Color. Here, we use a realist/essentialist paradigm [71] to propose that participants' recommendation that research mentors consider various demographic factors extending beyond race might potentially function as a tactic to indirectly avoid confronting issues related to racial inequities. For sample, a Latina woman explained:

> Color doesn't matter I suppose... Instead other things should be taken into account, like family situations or mental/physical issues and how the students themselves feel about their situation. Instead of focusing on one thing, look at everything as a bigger picture.

In addition, an Asian American woman explained, "I don't necessarily think it has to do with race, but each person they mentors personality and the way they would handle certain situations thats when the mentor should pose specific actions or support". These quotes highlight a perspective held by some students who believe that there may be other factors that hold greater importance than race and ethnicity in the context of mentorship for Students of Color. Although, as mentioned above, this may be a tactic used to indirectly circumvent addressing racial inequities.

### 3.3.5. Condescending

The third theme emphasizes that research mentors should not do anything in particular to support Students of Color because this will result in Students of Color feeling singled out or uncomfortable. Furthermore, participants stress that if research mentors do something in particular to support Students of Color, it will come across as conveying pity toward Students of Color. For example, one participant, an Asian American man, explained that when research mentors consider race, Students of Color will "take it more offensive or might take it as ingenuine". Another participant, a White woman, suggested that this action can rub students the wrong way, stating, "Treat them like everyone else because if you treat them like they are a person of color, that can rub them the wrong way". Lastly, a Black man explained, "I feel that, at least for me, being treated fairly and with respect is enough, that any extra behaviors would just make me feel different from my peers". These quotes reflect that some participants believe that it is not helpful for research mentors to take race into account because, through such actions, Students of Color may experience discomfort and perceive the mentor as patronizing or pitying them.

### 3.3.6. Favoring

The fourth theme centers on participants' belief that research mentors should avoid taking specific actions that could be perceived as preferential treatment, providing undue advantages, or favoritism toward Students of Color. For example, one participant, a White man, stated, "People should be treated the same. No one needs special treatment or advantages simply because of skin color". Similarly, another participant, a Latina woman, explained, "I think that research mentors shouldn't be favoring anyone's success over another". Lastly, a White man stated, "It should be promoted that everyone is treated with equal service, to prevent both prejudice and favoritism". These quotes show that some students believe mentors should be careful not to show favoritism or treat Students of Color differently from others. Moreso, these quotes emphasize that these students view efforts to achieve racial equity to mean providing advantages to Students of Color.

### 3.4. Ethnic Differences in Qualitative Themes

To address Research Question 2, the qualitative coding team conducted an analysis specifically focused on the qualitative responses provided by Black and Hispanic/Latinx participants to identify any unique or novel themes. In particular, we sought to examine differences beyond what was already listed in the qualitative codebook, such as touching on lived experiences with racial discrimination or variations in tone or valence. However, the qualitative analysis did not reveal any unique or novel themes specific to Black and Hispanic/Latinx participants. In the discussion section below, we provide an exploration of the potential methodological factors that might have contributed to the absence of variation in the qualitative responses. In addition, due to the categorical nature of our data, we conducted a chi-square analysis to determine whether coding membership differed significantly by racial group. However, chi-square analyses revealed no significant differences.

### 3.5. Quantitative

We used quantitative analyses to determine whether the ethnic-racial backgrounds of undergraduate students have an impact on their perception of whether research mentors should take race into account when mentoring students from marginalized backgrounds. More specifically, we used a $2 \times 2$ chi-square to compare White and Asian American participants with Black and Hispanic/Latinx participants in terms of whether they believe that research mentors should consider race when mentoring students from marginalized backgrounds (Hypothesis 1). Counter to expectations, the chi-square was not statistically significant (see Table 3).

We were also interested in capturing other demographic differences in regard to whether students believed research mentors should take race into account when mentoring students from marginalized racial backgrounds (Research Question 3). A chi-square

analysis revealed a statistically significant difference based on political party affiliation. Specifically, compared to all other political parties, Republicans were significantly less inclined to believe that research mentors should take race into account when mentoring students from marginalized backgrounds. The full results of the chi-square analyses are presented in Table 3. Furthermore, we employed an independent-sample *t*-test to assess whether participants who held the belief that mentors should not take race into account when mentoring students from marginalized backgrounds exhibited higher scores on racial colorblind ideologies. The results were statistically significant: Participants who endorsed the notion that mentors should not take race into account scored significantly higher on racial colorblind ideology. All other analyses testing for sociodemographic variation were not statistically significant.

## 4. Discussion

The primary aim of the current study was to examine undergraduates' perceptions of whether research mentors should consider race when mentoring students from marginalized racial backgrounds. We also attempted to examine whether participants who believe that research mentors should refrain from considering race when mentoring students from marginalized racial backgrounds would use colorblind ideologies to justify their stance. To achieve our objectives, we employed a mixed-methods approach. Our diverse sample predominantly expressed a preference for mentors who acknowledge race. Therefore, if previous research indicates widespread adoption of a colorblind mentoring approach by research mentors [2–4,11,12] and, at the same time, most surveyed students find such an approach to be unfavorable, it becomes a crucial focal point for intervention strategies. These findings add to the existing body of research by suggesting that a colorblind mentoring approach could lead to the attrition of Students of Color in STEM fields and highlight the need for mentors to recognize and address race in their mentoring relationships.

However, for some students, colorblind ideologies stand as a hurdle to believing that research mentors should take race into account when mentoring students from marginalized racial backgrounds. In the following sections, we provide a summary of our quantitative findings, highlight key qualitative themes identified during our coding process, and leverage these findings to propose a flexible intervention aimed at encouraging students and research mentors alike to redress colorblind ideologies and, consequently, endorse a more culturally sensitive mentoring approach.

### 4.1. Race-Conscious Mentoring Themes

After synthesizing the data within the *race-conscious mentoring* subgroup, we identified three overarching themes that encapsulate participants' justification for why they believe that research mentors should take race into account when mentoring students from marginalized backgrounds: (1) *culturally sensitive*, (2) *acknowledge adversity*, and (3) *negative behaviors.* First, *culturally sensitive* was characterized as emphasizing the importance of mentors providing a safe and encouraging environment for Students of Color and the need for research mentors to actively seek an understanding of diverse perspectives and the unique backgrounds of Students of Color. This is an encouraging finding, as a culturally sensitive mentoring style is believed to contrast the colorblind mentoring approach and foster a positive mentoring relationship [20]. Moreover, a culturally sensitive mentoring approach is associated with mentees having clearer academic and career goals and feeling more confident as researchers [37]. These findings highlight that a substantial number of undergraduate students endorse an inclusive mentoring approach and recognize that an effective research mentor not only provides guidance and support but also adopts a culturally sensitive mentoring approach.

The second overarching theme, *acknowledge adversity*, centers around the challenges faced by Students of Color, with numerous participants emphasizing the importance of mentors acknowledging these adversities when building a mentoring relationship. This is a promising finding as it suggests that undergraduate students find ignoring cultural dynam-

ics not to be an effective mentoring strategy. In fact, ignoring adversity can inadvertently worsen the situation and contribute to a decline in students' cognitive functioning [72]. Therefore, acknowledging adversity is essential for creating a supportive and inclusive mentoring relationship [73,74]. The third overarching theme, *negative behaviors,* addresses behaviors exhibited by mentors that may seem positive but can negatively impact the mentor-student relationship. This theme encompasses participants who advocate for research mentors to consider race while also emphasizing the need for such considerations to be done in an effective and non-harmful manner. Specifically, participants expressed concerns about mentors pretending to possess knowledge or expertise they do not actually have or assuming they have a complete understanding of the student's background when they may not. Participants raise a valid concern, as some academics may believe they are immune from cultural subjectivity or implicit bias [28,30,65]. This mindset could potentially lead to research mentors failing to recognize and address their own biases, resulting in unintentional discrimination and preservation of inequities in academia.

*4.2. Race-Ignoring Mentoring*

After synthesizing the data within the race-ignoring mentoring subgroup, we identified four overarching themes that encapsulate participants' justification for why they believe that research mentors should not take race into account when mentoring students from marginalized backgrounds: (1) *colorblind*, (2) *individualized*, (3) *condescending*, and (4) *favoring*.

The first overarching theme, *colorblind*, is comprised of two subthemes; these subthemes represent what Neville and colleagues [43] identified as the two categories of colorblindness. The first subtheme, *color evasion*, involves participants adopting a perspective of being "colorblind", wherein they claim not to see color. Although only one participant used a power evasion approach, we made the decision not to exclude this subtheme. Retaining this subtheme allows us to compare the frequency with which participants are drawing from either category of colorblindness. The second subtheme, *power evasion*, was used by the majority of participants in the race-ignoring mentoring group and is characterized as a lack of awareness regarding racial privilege and the institutional structures that contribute to inequality. This subtheme varies from the color evasion subtheme as participants avoided mentioning not seeing color and instead focused on racial equality (versus equity). The variation in the use of color evasion versus power evasion may be explained by considering that color evasion is associated with the refusal to recognize the concept of race itself, whereas power evasion is associated with the refusal to recognize the presence of institutional and systemic racial oppression [43]. It is possible that we observed a higher number of participants adopting a power evasion approach because these participants do not recognize or choose not to acknowledge the presence of institutional racism on college campuses. Taken together, the prevalence of the colorblind theme suggests that colorblindness is an influential factor shaping undergraduates' rationale for believing that research mentors should not take race into account.

The second overarching theme, *individualized*, highlights the importance of research mentors considering all aspects of a student's identity, including gender, sexual orientation, and socio-economic status when providing support for Students of Color. Although this theme may seem promising, it is possible that it serves as a strategy by participants to divert attention away from addressing racial inequities, allowing them to instead concentrate on other demographic variables. The third overarching theme, *condescending*, underscores the perception among some participants that research mentors should avoid doing anything in particular to support Students of Color because such actions will result in Students of Color feeling singled out or uncomfortable. This perspective may arise from the concern that explicitly addressing race could be misinterpreted as singling Students of Color out rather than as a genuine effort to promote racial equity and inclusivity within academic settings. These responses indicate a potential misunderstanding of the fundamental advantages of

promoting equity within the academic environment, underscoring the necessity for future efforts to rectify these misconceptions.

The fourth overarching theme, *favoring*, centers on participants' belief that research mentors should avoid taking specific actions that could be perceived as preferential treatment, providing undue advantages, or favoritism toward Students of Color. These participants may have inaccurately perceived racial equity to mean that Students of Color receive unfair advantages. Such concerns about favoritism reflect a commitment to upholding a meritocratic ethos within academia, wherein success is determined by individual talent and effort rather than external influences. Similar to the above, responses such as these indicate a potential misunderstanding of the fundamental advantages of promoting equity within the academic environment and call for future efforts to rectify these misconceptions.

### 4.3. Quantitative Findings

Quantitative analyses were employed to examine the influence of ethnic-racial backgrounds on undergraduate students' perceptions of whether research mentors should consider race when mentoring students from marginalized backgrounds (Hypothesis 1). However, the analyses did not yield any significant differences between racial categories. Interestingly, although only 30 percent of the sample believed that research mentors should not consider race, nearly half of the participants who held this view identified as Black or Hispanic and Latinx (46%). This finding might be explained by recent research [75,76], which revealed that some racial minority college students demonstrate resignation when facing racial discrimination because, for marginalized students, silence often emerges as the safer and sometimes the only viable option to cope with such situations. As a result, students may find themselves in a state of emotional detachment and desensitization toward racial discrimination. This emotional state may arise from the stigma and barriers surrounding the expression and validation of racialized experiences [77].

We also used quantitative analyses to explore other demographic differences in regard to whether students believed research mentors should take race into account when mentoring students from marginalized backgrounds (Research Question 3). The analysis revealed that, compared to all other political parties, students who identified as Republicans were significantly less inclined to believe that research mentors should take race into account when mentoring students from marginalized backgrounds. These findings are not unexpected, as previous research has shown that Republicans tend to believe that our society has made significant progress in achieving racial equity and, consequently, tend to be less supportive of initiatives aimed at achieving racial equity [78]. Furthermore, the analysis revealed that participants who endorsed the notion that mentors should not consider race scored significantly higher on racial colorblind ideology. These findings further support our qualitative findings, as they demonstrate that a vast majority of participants who opposed research mentors taking race into account relied on colorblind ideologies to justify their stance.

### 4.4. Implications and Recommendations for Retaining Students of Color in STEM

Collectively, our results indicate that the adoption of colorblind ideologies, particularly in the form of power evasion, poses a barrier to endorsing the notion that research mentors should take race into account when mentoring students from marginalized racial backgrounds. As noted above, although only 30 percent of the sample believed that research mentors should not consider race, nearly half of these participants identified as Black or Hispanic and Latinx. This finding highlights the importance of peer context within STEM learning. A peer environment wherein students from marginalized racial backgrounds recognize racial inequities offers them a platform to discuss the difficulties they encounter in the STEM field, confront biases, and collaborate to achieve success in the field [79,80]. Overall, our findings underscore the importance of addressing colorblind ideologies among both STEM students and research mentors. Achieving this will foster a learning atmosphere wherein STEM students from marginalized racial backgrounds can thrive.

In the following section, we present a flexible blueprint for a data-driven intervention designed to (1) encourage research mentors to endorse a colorblind mentoring approach and (2) convince "race-ignoring" students to redress colorblind ideologies and thus support a culturally sensitive mentoring style. Moreso, our primary objective is to utilize the outcomes derived in the current study to outline the fundamental elements of the intervention but allow flexibility in its actual implementation [81]. The flexibility enables the intervention to be applied effectively among participants from diverse demographic backgrounds and varying levels of readiness to resist colorblind ideologies.

Both our quantitative and qualitative findings demonstrate that the endorsement of colorblind ideologies serves as a barrier to believing that research mentors should take race into account. Moreover, many participants endorsed a power evasion approach, which is associated with the refusal to recognize the presence of institutional and systemic racial oppression [43]. Therefore, interventions should begin with an emphasis on promoting *critical thinking*. Critical thinking plays a pivotal role in developing the skills required to identify systems of oppression, recognize one's potential complicity in these systems, and understand how dominant cultural values have been accepted as truths and norms [82]. Critical questions will prompt participants to engage in discussions and acknowledge how racial oppression leads to unequal distribution of resources and significantly hampers access to educational opportunities and career advancement for students from marginalized racial backgrounds. For example, facilitators may pose questions to both students and research mentors, such as: What does it mean to hold colorblind views? Facilitators may proceed by asking students how colorblind perspectives and research mentoring practices impact their experiences in research settings and future career aspirations. Additionally, facilitators may inquire with research mentors about the impact of adopting a colorblind mentoring approach on Students of Color, their research experiences, and their prospects for remaining in the STEM field and pursuing a career in STEM.

Additionally, it is crucial for intervention facilitators to *break down the power dynamics* among individual participants. This is crucial because previous research has indicated that individuals with more social privilege tend to be more willing to voice their opinions within a group setting [83–85]. This could lead to White participants dominating the conversation and potentially silencing Participants of Color and preventing them from sharing their experiences with racial oppression. To mitigate this, intervention facilitators must foster a collective identity among intervention members by emphasizing *respect for all speakers' opinions*. Encouraging active participation from all members and acknowledging diverse opinions is crucial, as it promotes an inclusive environment [86]. Montero [87] recommends that interventions should not only encourage active participation from all group members but also foster an atmosphere of appreciation for contrasting opinions. However, it is important to note that although it is crucial to show appreciation for differing viewpoints, scholars [88] emphasize the importance of respectfully challenging opinions when necessary. For instance, if a participant were to share inaccurate information regarding students from marginalized racial backgrounds, it becomes crucial for the facilitator to correct any misconceptions by providing accurate information about the history and experiences of these students. Therefore, it is advisable that the intervention facilitator possesses a solid understanding of the background and experiences of students from marginalized racial backgrounds.

Engaging with a variety of viewpoints and opinions within the intervention will lead participants to a heightened *awareness of sociopolitical circumstances* [87,89]. This is important, considering our findings reveal that many participants believed research mentors should not consider race and hold misconceptions about the advantages of promoting equity within the academic environment. Through exposure to diverse perspectives, participants can integrate experiences different from their own, resulting in a deeper comprehension of the benefits associated with more equitable sociopolitical circumstances and the drawbacks of maintaining colorblind views [87,89]. After participants have developed the ability to recognize and critically analyze colorblind ideologies, participants will, ideally, show a

greater endorsement of culturally sensitive mentoring and understanding of the adversities experienced by students from marginalized racial backgrounds. In addition, once the intervention components outlined above have been implemented, participants may be better equipped to identify instances of oppression, marginalization, and privilege. Furthermore, research mentors may exhibit a greater willingness to embrace a culturally sensitive mentoring approach themselves. On a broader scale, students and research mentors alike may be inspired to initiate changes in both their personal lives and the lives of others [90,91].

*4.5. Limitations and Future Directions*

As with any study, there are limitations to the current study. Most notably, it is important to revisit Research Question 2, wherein we examined whether compared to Asian American and White students, Black and Hispanic and Latinx students differed in their beliefs about whether research mentors should take race into account when mentoring students from marginalized backgrounds. This is an intuitive question to investigate, considering Asian American and White individuals tend to endorse greater levels of colorblindness compared to members of other ethnic groups [32,46–48,50]. The absence of meaningful ethnic differences in the qualitative data was unexpected and could be attributed to several factors. One possibility is that the use of open-ended questions (compared to an interview) may have limited participants' capacity to fully elaborate on their reasons for supporting or opposing the consideration of race when mentoring students from marginalized backgrounds. Another explanation could be that our sample was too small to capture the nuances in the data. Future research should consider employing a different methodological approach, such as semi-structured interviews, to gain a more comprehensive understanding of how participants' ethnic and racial backgrounds influence their reasoning. This approach would provide participants with the opportunity to offer more detailed and nuanced responses, thereby enriching our comprehension of their perspectives and rationale.

Another potential limitation is associated with our qualitative prompt. The qualitative prompt was relatively broad and did not ask participants to consider contextual information such as the hypothetical mentor's racial background. It is possible that a more targeted prompt would have yielded more nuanced qualitative responses. For example, perhaps participants would provide varying responses depending on whether a research mentor is White or a Person of Color themselves. To this end, one idea for future research is to present participants with multiple qualitative prompts that include mentors from various racial backgrounds. This approach could help to uncover any potential differences in mentoring experiences based on the racial background of the mentor. By asking participants to consider the racial background of the mentor in their responses, researchers may gain a deeper understanding of how race can impact mentoring relationships in STEM fields.

Finally, it is essential to consider the potential influence of social desirability or response bias on participants' responses. Similar to points made by Hebert and colleagues [92], some participants may have tailored their responses toward the end of presenting a positive image. Therefore, it is a possibility that certain respondents provided insincere or exaggerated responses to survey items.

**5. Conclusions**

Prior research highlights the widespread adoption of a colorblind mentoring approach among research mentors [2–4,11,12]. Many academics are inclined to downplay racial privileges and instead endorse the idea that academia is a colorblind meritocracy in which everyone has an equal opportunity to be successful [93]. Our findings expand on this work by showing that although a minority of our sample endorse colorblindness, the majority of students do not favor a colorblind mentoring approach. This reinforces the notion that colorblind mentoring can result in students leaving STEM fields. To counter the colorblind mentoring approach, it is vital that research mentors be culturally sensitive,

meaning mentors are to be cognizant of their mentees' racial and ethnic backgrounds and adjust their mentorship strategies as appropriate [13,20].

This study contributes to the literature by providing an initial examination of how undergraduate students perceive and rationalize research mentoring practices. Furthermore, our findings uncover promising opportunities for future intervention efforts aimed at retaining Students of Color in the STEM field. We encourage others to expand upon our findings using methodological approaches that enable a more nuanced exploration of the data, aiming to gain a deeper comprehension of how undergraduate students perceive research mentoring practices.

**Author Contributions:** Conceptualization, K.D.V., D.R.B., L.D. and R.D.R.; methodology, K.D.V., D.R.B., L.D. and R.D.R.; formal analysis, K.D.V., D.R.B., L.D. and R.D.R.; investigation, K.D.V. and D.R.B.; resources, K.D.V., D.R.B. and R.D.R.; data curation, K.D.V. and D.R.B.; writing—original draft preparation, K.D.V. and D.R.B.; writing—review and editing, K.D.V., D.R.B., L.D. and R.D.R.; visualization, K.D.V.; supervision, R.D.R.; project administration, K.D.V.; funding acquisition, R.D.R. All authors have read and agreed to the published version of the manuscript.

**Funding:** This research received no external funding.

**Institutional Review Board Statement:** The study was conducted in accordance with the Declaration of and approved by the Ethics Committee of removed for peer review.

**Informed Consent Statement:** Informed consent was obtained from all subjects.

**Data Availability Statement:** The raw data supporting the conclusions of this article will be made available by the authors without undue reservation.

**Acknowledgments:** We are thankful for Jasmin Barajas and Stefany Flores Martinez help in analyzing the qualitative data.

**Conflicts of Interest:** The authors declare no conflicts of interest.

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
