# Peer review of "“Research Mentors Should Support Students of Color by Taking an Extra Step.” Undergraduates’ Reasoning about Race and STEM Research Mentorship"

_education, doi:10.3390/educsci14020162_

Round 1
Reviewer 1 Report
Comments and Suggestions for Authors The research is about how academia deals with racial inequalities and whether the idea of not seeing color in education affects our ability to confront racism. I find this research really interesting, and I think it's fresh and original. It could be published in the future. But I do have some suggestions for improvement before that happens. 1. The document lacks recent citations and should include some more up-to-date references.The references do not follow the journal referencing style, as some journal names are written in lowercase. 2. In the introduction section, it would be helpful to include a reference or a brief explanation of colorblind ideologies for the benefit of novice researchers or those not familiar with the topic. 3. Table 2 is not displaying correctly. Please replace it with a TIFF format image or recreate the table.4. Table 2 is not displayed correctly. Please replace it with a TIFF format image or recreate the table.
5. The research methodology is effectively explained, and the rationale for a mixed-method approach is sound. In the thematic analysis, it would be beneficial to include a greater number of excerpts (beyond just one or two) to gain a more comprehensive understanding of how participants have reacted to each theme in various ways, especially concerning subtypes of the "Race-Ignoring Mentoring Themes."
6. I couldn't locate Table 3 in the document. It would be helpful to clarify why the chi-square test was chosen over the more robust t-test and provide a justification for this choice.
7. Additionally, consider including a separate section titled "Directions for Future Research" to provide guidance for future studies.
Author Response
- The document lacks recent citations and should include some more up-to-date references.
- Thank you for suggesting we add more up-to-date references. We have added the following references to our literature review.
- Whitley, B.E.; Luttrell, A.; Schultz, T. The Measurement of Racial Colorblindness. Personality & Social Psychology Bulletin 2023, 49, 1531–1551, doi:10.1177/01461672221103414.
- Russo‐Tait, T. Color‐blind or racially conscious? How college science faculty make sense of racial/ethnic underrepresentation in STEM. Journal Of Research in Science Teaching 2022, 59, 1822–1852, doi:10.1002.
- King, G.P.; Russo-Tait, T.; Andrews, T.C. Evading Race: STEM Faculty Struggle to Acknowledge Racialized Classroom Events. CBE Life Sciences Education 2023, 22, ar14–ar14, doi:10.1187/cbe.22-06-0104.
- Callwood, K.A.; Marissa, W.; Rose, H.; Temis, T. Acknowledging and Supplanting White Supremacy Culture in Science Communication and STEM: The Role of Science Communication Trainers. Frontiers in communication 2022, 7, doi:10.3389/fcomm.2022.787750.trainers. Frontiers in Communication, 7, 35.
- The references do not follow the journal referencing style, as some journal names are written in lowercase.
- Good catch! We have fixed the journal names so that they are no longer written in lowercase.
- In the introduction section, it would be helpful to include a reference or a brief explanation of colorblind ideologies for the benefit of novice researchers or those not familiar with the topic.
- Great idea! We have added an explanation of colorblind ideologies (page 1) towards the top of the document and expanded on our original explanation of colorblind ideologies (page 3).
- Table 2 is not displayed correctly. Please replace it with a TIFF format image or recreate the table.
- Thank you for flagging this. We have replaced all of our tables with a TIFF format image.
- In the thematic analysis, it would be beneficial to include a greater number of excerpts (beyond just one or two) to gain a more comprehensive understanding of how participants have reacted to each theme in various ways, especially concerning subtypes of the "Race-Ignoring Mentoring Themes."
- We appreciate the encouragement to add more qualitative excerpts. Therefore, we have added an additional excerpt to the following qualitative theme write-ups:
- Culturally sensitive
- Acknowledge diversity
- Power evasion
- Individualized.
- Condescending
- Favoring
- I couldn't locate Table 3 in the document. It would be helpful to clarify why the chi-square test was chosen over the more robust t-test and provide a justification for this choice.
- Thank you for flagging this! We believe something went wrong with the file format in the original submission. Table 3 can now be found on page 9.
- Thank you for your inquiry regarding our statistical analyses. We opted for a chi-square test instead of a t-test due to the categorical nature of our data. We have added a rationale for employing chi-squares, which can be found on page 14.
- Additionally, consider including a separate section titled "Directions for Future Research" to provide guidance for future studies.
- Thank you for your insightful suggestion. We agree that we need to elaborate on future directions and have added more detail on page 21. Our directions for future research are included alongside our limitations (found in the “Limitations and future directions section”) because we believe they complement one another.
Reviewer 2 Report
Comments and Suggestions for Authors
Congratulation, your research achievements reflects your novel approach. I recommend the acceptance of your paper in its current format.
Author Response
Reviewer 2 did not provide comments.
Reviewer 3 Report
Comments and Suggestions for Authors
Dear editor
Dear authors,
This paper examines undergraduates' perceptions of whether STEM research mentors should consider race when mentoring students from marginalized backgrounds. I believe that this interesting paper has significant contribution in education field. below authors can see my comment to improve the quality of this manuscript.
1. What is the relationship between Racial inequities and STEM research mentor and not in general mentor?
2. Line 11, author need to write what is STEM?
3. Introduction section: the authors not provide contribution of their study, novelty research and the explanation of the structure of paper in the last paragraph in introduction section
4. Literature review section is missing
5. Again, author didn’t mention about STEM in education but more focus on sensitive mentoring approach.
Overall manuscript presents an intriguing idea with a well-organized structure and in-depth discussions. However, in the context of STEM (Science, Technology, Engineering, and Mathematics), it would be beneficial to highlight the explicit links or implications of authors work within these fields.
Consider elucidating the potential applications or contributions authors research could make to any scientific advancements, technological innovations, engineering methodologies, or mathematical frameworks. Emphasizing how authors findings intersect with or impact STEM domains could enhance the manuscript's relevance and appeal to a wider audience within these disciplines
Author Response
- What is the relationship between Racial inequities and STEM research mentor and not in general mentor?
- This is a great question! It is something we have discussed at length within our research team. To better explain why we have chosen to focus on racial inequities in STEM mentoring, we have restructured the first paragraph of the introduction to focus more specifically on STEM fields. In addition, on page 4, we have added a paragraph outlining the context of racial inequities within STEM. We believe this paragraph will provide a greater rationale for examining racial inequities specific to STEM.
- Line 11, author need to write what is STEM?
- Great idea! On line 11 and 12, we now write out what the STEM acronym stands for.
- Literature review section is missing.
- While preparing the manuscript, I was guided by the journal template, which does not include a literature review subheading. However, the literature review content begins in section 1.1. If the editorial team deems it necessary to include a subheader for the literature review, I am more than willing to make the adjustment. However, there is some uncertainty regarding the appropriateness of this addition, considering the formatting standards of the journal.
- Again, author didn’t mention about STEM in education but more focus on sensitive mentoring approach. It would be beneficial to highlight the explicit links or implications of authors work within these fields.
- We agree that it would be helpful to elaborate on our reasoning for focusing on racial inequities in the STEM fields. On page 4, we have added a paragraph outlining the context of racial inequities within STEM. We believe this paragraph will provide a stronger rationale for examining racial inequities specific to STEM.
- Consider elucidating the potential applications or contributions authors research could make to any scientific advancements, technological innovations, engineering methodologies, or mathematical frameworks. Emphasizing how authors findings intersect with or impact STEM domains could enhance the manuscript's relevance and appeal to a wider audience within these disciplines.
- Thank you for encouraging us to elaborate on how our findings impact STEM domains. We believe this will make our manuscript stronger. We have added a section on page 18 discussing the importance of peer context in STEM learning and a corresponding citation (i.e., Ong et al., 2018) indicating the importance of “safe spaces” within the STEM peer context.
Round 2
Reviewer 3 Report
Comments and Suggestions for Authors
Dear Editor
Dear Author
I have read this paper, this paper ready to publish.
well done